# Prognostic Significance of Tumor Regression Rate during Concurrent Chemoradiotherapy in Locally Advanced Cervix Cancer: Analysis by Radiation Phase and Histologic Type

**DOI:** 10.3390/jcm9113471

**Published:** 2020-10-28

**Authors:** Jun-Hyeok Kang, Won Kyung Cho, Hie Jun Yeo, Soo Young Jeong, Joseph J. Noh, Jung In Shim, Yoo-Young Lee, Tae-Joong Kim, Jeong-Won Lee, Byoung-Gie Kim, Duk-Soo Bae, Won Park, Chel Hun Choi

**Affiliations:** 1Department of Obstetrics and Gynecology, Samsung Medical Center, Sungkyunkwan University School of Medicine, Seoul 06351, Korea; junhyeok.kang@samsung.com (J.-H.K.); hj1118.yeo@samsung.com (H.J.Y.); sy1130.jeong@samsung.com (S.Y.J.); joseph.noh@samsung.com (J.J.N.); jiin.shim@samsung.com (J.I.S.); yooyoung.lee@samsung.com (Y.-Y.L.); tj28.kim@samsung.com (T.-J.K.); garden.lee@samsung.com (J.-W.L.); bksong.kim@samsung.com (B.-G.K.); ds123.bae@samsung.com (D.-S.B.); 2Department of Radiation Oncology, Samsung Medical Center, Sungkyunkwan University School of Medicine, Seoul 06351, Korea; wklove.cho@samsung.com

**Keywords:** histologic subtype, locally advanced cervical cancer, concurrent chemoradiotherapy, regression, survival

## Abstract

This study aimed to evaluate the prognostic significance of tumor regression rate according to radiation phase and histologic subtype in patients with locally advanced cervical cancer (LACC) treated with chemoradiation. We retrospectively reviewed the medical records of 398 patients with FIGO stage IIB-IVA cervical cancer treated with concurrent chemoradiotherapy (CCRT) between 2001 and 2019. Tumor response was assessed using serial magnetic resonance imaging (MRI) at three time points: pre-treatment, post-external beam radiotherapy (EBRT), and post-intracavitary radiotherapy (ICR). Tumor regression pattern according to histologic subtype and radiation phase (EBRT and ICR) was evaluated. Overall survival (OS) and progression-free survival (PFS) were the primary outcomes. Of 398 patients, 44 patients had adenocarcinoma/adenosquamous carcinoma (AC/ASC) and 354 patients had squamous cell carcinoma (SCC). AC/ASC was associated with significantly worse PFS and OS than SCC (*p* < 0.001). AC/ASC had a relatively poorer regression rate in response to EBRT than SCC (*p* < 0.001), whereas there was no significant difference in overall tumor regression rate after completion of RT (EBRT and ICR) between the two histologic subtypes. Multivariable analysis demonstrated AC/ASC histology to be an independent prognostic factor of decreased PFS and OS. Moreover, tumor regression rate after completion of EBRT (post-EBRT tumor regression rate (EBRT_regression_ ≤ 26%) and proportion of tumor regression during EBRT to overall tumor regression (EBRT_proportion_ ≤ 40%) were independent predictors of poor survival in patients with LACC. Tumor regression pattern of LACC in response to CCRT differs according to histologic subtype. AC/ASC histology and poor tumor response to EBRT are independent prognostic factors for worse survival in patients with LACC. Further studies are needed to develop a CCRT protocol that is specialized for patients with AC/ASC.

## 1. Introduction

Cervical cancer is the third most common cancer in women worldwide, accounting for 9% of total new female cancers [1]. Although the prognosis of locally advanced disease is poor, the introduction of concurrent chemotherapy has improved survival of these patients compared with radiation therapy (RT) alone due to the synergistic interaction between chemotherapy and RT [2,3,4]. Concurrent chemoradiation therapy (CCRT) is therefore used as the standard treatment for locally advanced cervical cancer (LACC) [3]. However, RT, composed of external beam radiotherapy (EBRT) and intracavitary radiotherapy (ICR), still plays a central role in treating advanced stage cervical cancer compared to chemotherapy or surgery [5]. The external beam portion of treatment encompasses treats the pelvic lymph nodes, parametria, the primary tumor, and microscopic disease. The addition of brachytherapy serves to boost the primary tumor, and improves disease control and survival [6,7].

Response to RT is an important prognostic factor to predict survival outcomes [8,9,10,11]. Various clinical factors such as histologic subtype, pre-treatment tumor size, and the use of chemotherapy can affect responsiveness to RT [12,13]. Evaluation of tumor response during and at the end of RT is controversial; it is not yet clear at which time point evaluation of the RT response is best for prediction of survival. According to recent studies by Mayr et al. [14] and Wang et al. [11], tumor response rate measured in the middle of the entire RT process provides greater prognostic information than residual tumor status after completion of RT. It would be valuable to be able to predict overall survival outcomes based on early assessment of tumor response during RT. This information could also be used to guide early interventions for patients with LACC at high risk of treatment failure. However, most previous studies focused on pre-treatment tumor burden or post-treatment tumor response as prognostic factors, not intermediate tumor response. In addition, despite the fact that histological differences could affect the response rate to the RT phases (EBRT and ICR) as well as overall response rate after completion of RT [15,16,17], response of the tumor to RT based on PT phases and tumor histology has not been investigated previously.

Therefore, our purpose in this study was to investigate tumor regression rate according to histologic subtype and RT phase (EBRT and ICR), and to evaluate the prognostic significance of histologic subtype and responsiveness to EBRT in patients with LACC.

## 2. Materials and Methods

### 2.1. Inclusion and Exclusion Criteria 

We retrospectively analyzed the medical records of patients who underwent CCRT from 2001 to 2019 at Samsung Medical Center, Seoul, Korea. This study was approved by the Institutional Review Board (IRB) of Sungkyunkwan University of Korea (Ethical approval code: SMC2020-07-091-001). Patients who met the following criteria were eligible for inclusion in this study: (1) diagnosis of stage IIB—IVA cervical cancer based on the International Federation of Gynecology and Obstetrics (FIGO) 2014 staging classification, (2) histologically confirmed squamous cell carcinoma (SCC), adenocarcinoma (AC) or adenosquamous carcinoma (ASC), and (3) CCRT treatment. We excluded patients if they met one or more of the following criteria: (1) failure to complete the planned RT schedule, (2) incomplete medical records, or (3) treated with neo-adjuvant chemotherapy or surgery prior to initiation of RT.

### 2.2. Treatment

All patients were treated with a combination of external beam radiotherapy (EBRT) and high-dose-rate (HDR) intracavitary brachytherapy (ICR). EBRT was delivered to the whole pelvis five times per week with a 10 MV photon beam at a daily dose of 1.8 gray (Gy), for a total dose of 50.4 Gy. Four-field box technique using anteroposterior/posteroanterior and two lateral fields was used for EBRT. HDR ICR was initiated after an EBRT dose of 45 Gy and delivered three times a week in six fractions with a fractional dose of 4 Gy. For ICR planning, 2-dimensional technique using A point was performed in 231 patients (58.0%) and 3-dimensional planning based on computed tomography (CT) and 18F-fluorodeoxyglucose (FDG) positron emission tomography (PET)/CT simulation was performed in 2 (0.5%) and 165 patients (41.5%), respectively. The details of the PET/CT-based ICR are described in our previous studies [18,19]. EBRT was accompanied by concurrent chemotherapy comprising six cycles of weekly cisplatin (30 mg/mm^2^).

### 2.3. Assessment of Treatment Outcomes

Serial MRI examinations were performed at three time points to evaluate early treatment response: at the start of RT (pre-RT), at the fourth week of RT (post-EBRT), and 1 month after completion of RT (post-ICR). Tumor size, defined as the maximum diameter of the tumor measured using electronic calipers on MRI, was obtained for each time point: pre-RT tumor size (D1), post-EBRT tumor size (D2), and post-ICR tumor size (D3). Tumor size regression rates (%) were calculated as follows: post-EBRT regression rate (EBRT_regression_) = (D1–D2)/D1; post-ICR regression rate (ICR_regression_) = (D2–D3)/D2; and overall regression rate (RT_regression_) = (D1–D3)/D1. The ratio of the tumor size reduction after EBRT to overall tumor size reduction after RT (EBRT_proportion_) was calculated using the following formula: (D1 – D2)/(D1 – D3). Cut-off values for EBRT _regression_, EBRT_proportion_, and RT_regression_ were identified in a stepwise manner using 1% increments. Each parameter threshold was correlated with survival outcome. The most discriminating threshold values for EBRT_regression_, EBRT_proportion_, and RT_regression_ were 26%, 40%, and 92%, respectively. Patients were classified based on the 26% cut-off value of EBRT_regression_ as good EBRT responders (EBRT_regression_ > 26%) or poor EBRT responders (EBRT_regression_ ≤ 26%). Patients were also classified as more EBRT responders (EBRT_proportion_ > 40%) or more ICR responders (EBRT_proportion_ ≤ 40%) based on a 40% cut-off value for EBRT_proportion_. Patients were classified as good RT responders (RT_regression_ > 92%) or poor RT responders (RT_regression_ ≤ 92%) according to the 92% cut-off value of RT_regression_. Response to treatment was assessed according to the Response Evaluation Criteria in Solid Tumors (RECIST). Complete response (CR) was defined as the disappearance of the target lesion and the absence of a new lesion on two consecutive assessments. Partial response (PR) was defined as at least a 30% reduction in the sum of the longest dimension of the target lesion. Progressive disease (PD) was defined as at least a 20% increase in the sum of the longest dimension of the target lesion or the development of new lesion. Patients with a response that did not meet any of the criteria described above were considered to have stable disease (SD). Progression-free survival (PFS) was defined as the date of the first treatment until progression, recurrence, death, or follow-up loss, whichever occurred first. Overall survival (OS) was defined as the interval from the day of first treatment to the date of death or last contact.

### 2.4. Statistical Analysis 

Normality of the data was assessed with the Shapiro–Wilk test. Means ± standard deviations (SD) are reported for data with a normal distribution, while medians (interquartile ranges, IQR) are reported for data with a non-normal distribution. Frequency distributions of categorical variables for the four stage groups were compared using the chi-square test or Fisher’s exact test. Quantitative variables were compared using one-way analysis of variance (ANOVA) as a parametric test or the Kruskal–Wallis test as a non-parametric test. Survival curves were calculated according to the Kaplan–Meier method with the log-rank test. The Cox proportional hazards model was used for multivariate analysis to assess the independence of different prognostic factors. *p* < 0.05 was considered to indicate statistical significance. Statistical analyses were performed using SPSS software (IBM SPSS statics for Windows, Version 25.0. Armonk, NY, USA: IBM Corp.).

## 3. Results

Three hundred ninety-eight patients were included in this retrospective study (Figure 1). Of these, 354 patients (88.9%) had SCC and 44 (11%) patients had AC/ASC. Clinicopathological characteristics of the patients are summarized in Table 1. Age at diagnosis, FIGO stage, pre-treatment tumor size, and duration of RT were similar between the two histological subtypes (all *p* > 0.05).

Tumor size regression rates during RT according to histologic subtype are shown in Table 2. The mean EBRT_regression_ value was significantly lower in patients with AC/ASC (*n* = 44) than in patients with SCC (*n* = 354) (53.6% vs. 32.8%, *p* < 0.001). Radiological CR ratio of patients with AC/ASC on post-EBRT MRI was significantly lower than that of SCC patients (13.6% (48/354) vs. 6.8% (3/44), *p* < 0.001). ICR _regression_ was somewhat higher in patients with AC/ASC than in patients with SCC (73.6% vs. 77.9%), but this difference was not statistically significant. CR rate after ICR was comparable between the two histologic subtypes (53.3% (163/354) vs. 46.3% (19/44)). When we compared overall response to RT, there was no significant difference in size regression rate (RT_regression_, 87% vs. 83.2%) or CR rate (59.6% (211/354) vs. 50.0% (22/44)) between SCC and AC/ASC. EBRT resulted in a 60.5% overall tumor size regression in patients with SCC, whereas only 30.9% of the overall tumor size regression was induced by EBRT in patients with AC/ASC (*p* < 0.001). Significantly more patients with SCC showed tumor size regression of more than 26% after EBRT (good EBRT responders) than patients with AC/ASC (85.6% (303/354) vs. 52.3% (23/44), *p* < 0.001). The number of patients whose tumor size regression upon EBRT accounted for more than 40% of the overall size regression (more EBRT responders) was also significantly higher in those patients with SCC than in those with AC/ASC (78.2% (277/354) vs. 38.6% (17/44), *p* < 0.001). However, there was no significant difference in the number of patients who showed an overall tumor size regression of greater than 92% after completion of RT (good RT responders) between the two histologic subtypes.

In survival analysis, AC/ASC patients had a significantly shorter PFS and OS than SCC patients (Figure 2A and Appendix A). Estimated five-year OS rates of patients with SCC and AC/ASC were 68.1% and 44.2%, respectively. Survival differences according to EBRT_regression_ are shown in Figure 2B and Appendix A. Poor EBRT responders (EBRT_regression_ ≤ 26%) showed poorer survival than good EBRT responders (EBRT_regression_ > 26%) (PFS and OS, *p* < 0.001). More ICR responders (EBRT_proportion_ ≤ 40%) showed worse PFS (*p* = 0.003, Figure 2C) and OS (*p* < 0.001, Appendix A) than more EBRT responders (EBRT_proportion_ >40%). However, overall size regression rate (RT_regression_ cut-off value of 92%) was not associated with prognosis (Appendix A). Incorporating histology and responsiveness to RT, as shown in Figure 2D–F and Appendix A, AC/ASC patients with poor EBRT response (EBRT_regression_ ≤ 26%), more ICR response (EBRT_proportion_ ≤ 40%), and poor overall response (RT_regression_ ≤ 92%) were associated with significantly shorter PFS (*p* < 0.001) and OS (*p* < 0.001). Furthermore, patients who had a relatively better response to EBRT had a favorable survival outcome regardless of achieving CR after completion of RT (*p* < 0.05, Figure 3).

In a multivariate analysis using Cox proportional hazard modeling (Table 3), histologic subtype was a significant independent prognostic factor for PFS (SCC vs. AC/ASC, HR: 2.37, 95% CI 1.45–3.89, *p* = 0.001) and OS (SCC vs. AC/ASC, HR: 1.91, 95% CI 1.18–3.09, *p* = 0.009). Pre-treatment tumor size > 4 cm was also independently associated with survival. EBRT_regression_ was identified as a significant prognostic factor for survival (HR: 2.16 for PFS and HR: 2.53 for OS, *p* = 0.001 and *p* = 0.008). Moreover, EBRT_proportion_ was found to be an important prognostic factor for PFS (HR: 2.43, *p* = 0.031) and OS (HR: 2.59, *p* = 0.015). However, overall regression rate in response to RT (RT_regression_) was not associated with survival. The results of univariate analysis for theses clinical factors were presented in Appendix A.

## 4. Discussion

This study assessed tumor regression rate according to histologic subtype and RT phase during CCRT, and evaluated the prognostic significance of histologic subtype and responsiveness to EBRT. We found that patients with AC/ASC showed a significantly poorer response to EBRT than those with SCC. Histologic subtype and responsiveness to EBRT were independent prognostic factor for survival in patients with LACC.

CCRT is considered the standard treatment modality for patients with LACC. Even though survival outcomes have improved significantly since the introduction of concurrent chemotherapy [3], RT still plays a central role compared to surgery or chemotherapy when managing advanced stage cervical cancer patients. Conflicting results have been reported for AC/ASC histology regarding their response to therapy and prognosis compared to SCC. Although some studies have reported that an AC/ASC histology does not affect survival outcomes [20], the majority of studies have found that prognosis varies according to histological subtype with AC/ASC associated with a poorer prognosis than SCC. According to large retrospective study based on National Cancer Institute’s Surveillance, Epidemiology, and End Results (SEER) [21], AC showed unfavorable prognosis in both early stage and advanced stage compared with SCC (HR 1.39 and HR 1.21, respectively). Rose et al. [2] reported that AC histology was related to worse survival outcomes than SCC when treated with RT alone, but such differences in survival disappeared when tumors were treated with CCRT. Yokoi et al. [22] and Chen et al. [23] reported that AC/ASC patients had a poorer OS than those with SCC regardless of treatment modality (CCRT or RT alone). In accordance with previous studies, we found that patients with AC/ASC of the cervix treated with CCRT had inferior survival outcomes than those with SCC (*p* < 0.001), and that AC/ASC histology was an independent prognostic factor for a poor PFS and OS (HR 3.28 for PFS and HR 2.06 for OS).

Various hypotheses have been proposed to explain the worse prognosis in patients with an AC histology than those with an SCC histology, including radio-resistance. Some previous studies suggested possible mechanisms for radio-resistance in AC including a slow cell cycle and overexpression of villin 1 or cyclooxyengase-2 (COX-2) compared with SCC [24,25,26]. Incomplete tumor regression after completion of RT is considered an important prognostic factor for poor survival. According to a study that evaluated the incidence of residual tumor after RT for FIGO stage IB cervical cancer [27], patients with AC/ASC had a higher incidence of residual disease than patients with SCC: 91% vs. 48% (*p* = 0.001). Poujade et al. [17] also reported that 67% of stage IB-IIIB cervical AC patients had a pathologic residual tumor after CCRT. Couvreur et al. [16] revealed that AC patients treated with CCRT show significantly more pathologic residual disease than SCC patients (91% vs. 57%, *p* = 0.027). In our study cohort, however, incidence of residual tumor after completion of RT was 50.0% and 40.4% for AC and SCC, respectively (*p* = 0.474). Moreover, there was no statistically significant difference in CR rate between AC/ASC and SCC in our study. In particular, the CR rate of patients with AC/ASC was markedly higher than that reported in previous studies. The exact reason for this discrepancy is unknown, but may be due to several factors. First, we evaluated tumor regression based on radiologic response, not pathologic response. Furthermore, chemotherapy, which acts as radio-sensitizer, may have a more important effect on survival in AC/ASC patients than SCC patients. We found that tumors with an AC/ASC histology showed a comparable overall tumor size regression rate after primary treatment to those with an SCC histology, but worse long-term survival. This indicates that the disease progression pattern of AC/ASC is different from that of SCC, and that adjuvant treatment strategies after primary CCRT are important in AC/ASC patients. One possible treatment strategy is to use neo-adjuvant or adjuvant chemotherapy in combination with CCRT to eradicate micrometastases in AC/ASC patients. A randomized control trial study of 880 LACC patients with AC/ASC [28] revealed that patients who received CCRT with adjuvant chemotherapy had significantly longer DFS, OS, and local control than those who received CCRT alone (*p* < 0.05). Another possible scenario is salvage hysterectomy. In a recent study that evaluated the effect of adjuvant hysterectomy on survival of LACC patients with AC/ASC [29], the adjuvant hysterectomy after CCRT group showed better three-years PFS (68.1% vs. 45.2%, *p* = 0.002), three-year OS (87.9% vs. 67%, *p* = 0.002), and local control than the CCRT-only group.

We also analyzed the pattern of tumor size regression according to RT phase (EBRT and ICR, Appendix A), and assessed the prognostic value of early tumor response evaluation in terms of predicting survival. AC/ASC patients had a relatively poorer response to EBRT than those with SCC (EBRT_regression_, 53.6% vs. 32.8%, *p* < 0.001), but a comparable overall size regression rate (RT_regression_, 87% vs. 83.2%, *p* = 0.222). EBRT also contributed more to the overall reduction in size of SCC tumors than AC/ASC tumors (EBRT_proportion_, 60.5% vs. 30.9%, *p* < 0.001). Furthermore, tumor responsiveness to EBRT, including EBRT_regression_ ≤ 26% (HR 2.16 and HR 2.43, PFS and OS, respectively) and EBRT_proportion_ ≤ 40% (HR 2.31 and HR 2.69, PFS and OS, respectively), appeared to have greater prognostic value than overall regression rate after completion of RT. Hatano et al. [30] reported that patients with a tumor size regression rate over 70% during RT (at 30 Gy of EBRT) had good local control. Another study assessed the tumor volume regression rate and prognostic significance of EBRT response in 84 patients [15], and reported that tumor volume (tumor volume regression rate) after EBRT was 5.7 cc (90.8%) and 3.3 cc (87%) for AC and SCC, respectively (P value not shown). In addition, they found that tumor volume after EBRT and histologic subtype were independent prognostic factors for survival, and an absolute tumor volume after EBRT ≥ 7.5 cc was significantly associated with survival. Wang et al. [11] also reported that a tumor regression rate ≤80% at 4 to 5 weeks after initiation of RT was a strong predictor of a poor prognosis. Similar to previous studies, our results re-emphasize the prognostic value of responsiveness to EBRT in predicting survival. However, previous analyses did not consider histologic subtype or RT response simultaneously. When we considered both histology and responsiveness to EBRT simultaneously, we found that AC/ASC patients with a poor response to EBRT (poor EBRT responders or more ICR responders) had significantly poorer survival outcomes with SCC with good response to EBRT (good-EBRT responder or more-EBRT responder) (*p* < 0.001). However, overall regression rate had no prognostic value compared with histologic type and EBRT responsiveness. It is not clear why histologic differences affect EBRT response and why the tumor regression rate during RT was a more significant prognostic indicator than overall response. However, the results of the current study indicate that there is a need for a more effective RT protocol that considers histologic subtype, as well as the need for more appropriate tumor response evaluation timing to predict prognosis. The total radiation dose of this study is lower than the recommended radiation dose (80–90 Gy equivalent dose at 2 Gy [EQD2]) in international guidelines [31]. We had adopted the current treatment scheme based on the results of Japanese group which demonstrated the favorable outcomes in Asian patients with lower radiation dose recommended in USA and Europe [32,33,34] and our group also reported the favorable results of with this treatment scheme [19]. The necessity of higher ICR dose to control larger tumors compared to the small tumors is well known in cervical cancer; however, the effect of stratified radiation dose according to the different histologic types has not been evaluated [35]. We carefully consider increasing ICR dose in patients with AC/ASC histology who showed better response to ICR than EBRT in this study and exploring the benefit of higher dose in AC/ASC on treatment outcomes.

The main strength of this study was to evaluate tumor response according to RT phase and histologic subtype. However, our study also had some limitations. First, it was a retrospective chart review study. Second, the number of patients with AC/ASC histology was relatively small. Third, there was no consideration of tumor grade, considered as one of the prognostic factors, and there was no pathologic review. Furthermore, tumor size was assessed using only the maximum diameter of the tumor, not tumor volume. However, given that in gynecologic oncology practice tumor response to treatment is evaluated by measuring the diameter of target lesions, we think that it is more useful method than tumor volume measurement requiring more complex calculation. 

## 5. Conclusions

The results of the current study reaffirm that patients with AC/ASC of the cervix experience significantly worse survival than those with SCC. We also revealed that tumor regression pattern differed between AC/ASC and SCC during CCRT and that tumor responsiveness to EBRT was an independent prognostic factor of PFS and OS. Although current guidelines for cervical cancer recommend the same CCRT protocol regardless of histologic subtype, further clinical studies are needed to develop a CCRT protocol for LACC patients with AC/ASC to improve their survival. 

## Figures and Tables

**Figure 1 jcm-09-03471-f001:**
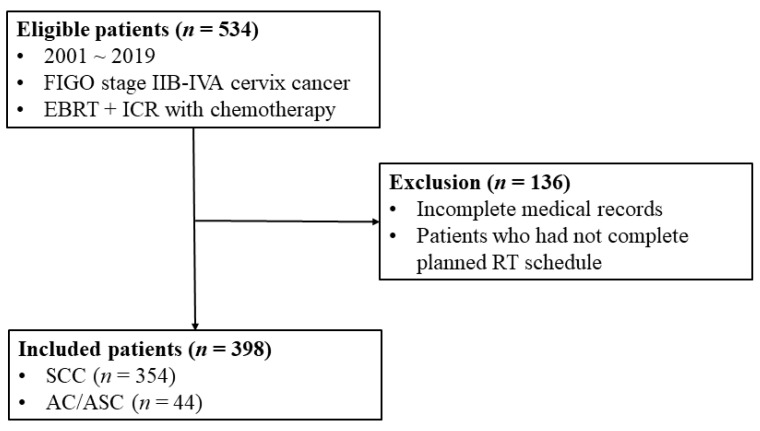
Inclusion and exclusion criteria for the study population. FIGO, International Federation of Gynecology and Obstetrics; EBRT, external beam radiotherapy; ICR, intracavitary radiotherapy; RT, radiation therapy; SCC, squamous cell carcinoma; AC, adenocarcinoma; ASC, adenosquamous carcinoma.

**Figure 2 jcm-09-03471-f002:**
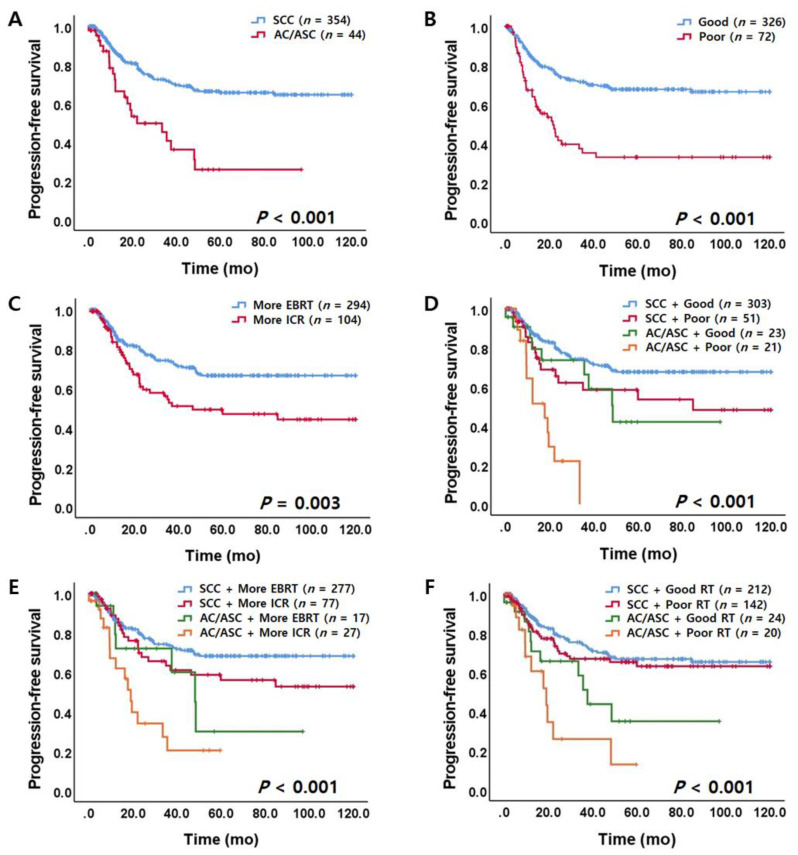
Progression-free survival (PFS) according to histologic subtype and responsiveness to RT. (**A**) PFS according to histologic subtype. (**B**) PFS according to post-external beam radiotherapy (EBRT)_regression_. (**C**) PFS according to EBRT_proportion_. (**D**) PFS according to histologic subtype and EBRT_regression_. (**E**) PFS according to histologic subtype and EBRT_proportion_. (**F**) PFS according to histologic subtype and RT_regression_. RT, radiation therapy; SCC, squamous cell carcinoma; AC, adenocarcinoma; ASC, adenosquamous carcinoma; EBRT, external beam radiotherapy; ICR, intracavitary brachytherapy; EBRT_regression_, post-EBRT tumor size regression rate; EBRT_proportion_, proportion of EBRT to overall size regression; RT _regression_, overall regression rate after completion of RT; Good, good EBRT responder (EBRT_regression_ > 26%); Poor, poor EBRT responder (EBRT_regression_ ≤ 26%); More EBRT, more EBRT responders (EBRT_proportion_ > 40%); More ICR, more ICR responders (EBRT_proportion_ ≤ 40%); Good RT, good RT responders (RT_regression_ > 92%); Poor RT, poor RT responders (RT_regression_ ≤ 92%).

**Figure 3 jcm-09-03471-f003:**
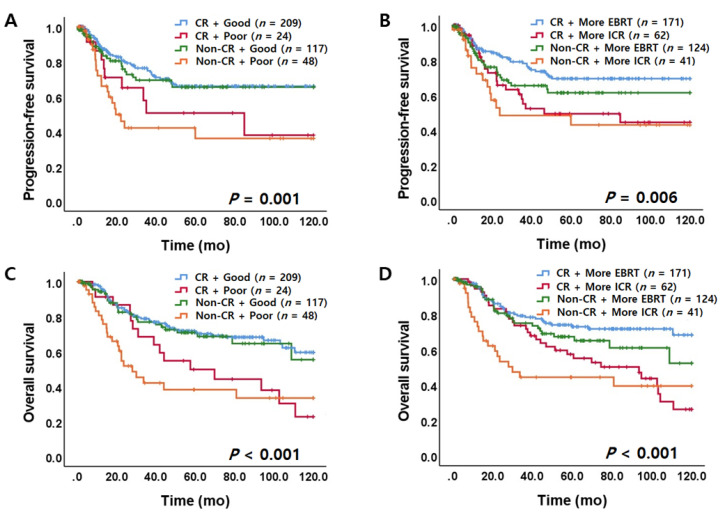
Long-term survival outcomes when incorporating early response and overall response to RT. (**A**) PFS according to RT_regression_ (CR or Non-CR) and EBRT_regression_. (**B**) PFS according to RT_regression_ (CR or Non-CR) and EBRT_proportion_. (**C**) OS according to RT_regression_ (CR or Non-CR) and EBRT_regression_. (**D**) OS according to RT_regression_ (CR or Non-CR) and EBRT_proportion_. RT, radiation therapy; PFS, progression-free survival; RT_regression_, overall regression rate after completion of RT; CR, complete remission; EBRT, external beam radiotherapy; ICR, intracavitary radiotherapy; EBRT_regression_, post-EBRT tumor size regression rate; EBRT_proportion_, proportion of EBRT to overall size regression; Good, good EBRT responder (EBRT_regression_ > 26%); Poor, poor EBRT responder (EBRT_regression_ ≤ 26%); More EBRT, more EBRT responders (EBRT _proportion_ > 40%); More ICR, more ICR responders (EBRT_proportion_ ≤ 40%).

**Table 1 jcm-09-03471-t001:** Clinicopathological characteristics of patients with SCC or AC/ASC.

Characteristics		Number of Patients (%)			
	.	All Patients (*n* = 398)	SCC (*n* = 354)	AC/SCC (*n* = 44)	*p*-Value
Age (years)	Mean ± SD	57.1 ± 11.9	57.4 ± 12.0	56.5 ± 12.2	0.615
	≤50	111 (27.9)	97 (27.4)	14 (31.8)	0.538
	>50	287 (72.1)	257 (72.6)	30 (68.2)	
Pretreatment Hb (g/dL)	Mean ± SD	11.5 ± 1.9	11.5 ± 1.8	11.9 ± 2.1	0.146
	<11	126 (31.6)	113 (31.9)	13 (29.5)	0.749
	≥11	272 (68.4)	241 (68.1)	31 (70.5)	
FIGO stage	IIB	232 (58.3)	206 (58.2)	26 (59.1)	0.877
	IIIA	19 (4.8)	16 (4.5)	3 (6.8)	
	IIIB	84 (21.1)	75 (21.2)	9 (20.5)	
	IIIC	20 (5.0)	19 (5.4)	1 (2.3)	
	IVA	43 (10.8)	38 (10.7)	5 (11.4)	
Tumor marker	SCC (ng/mL)	21.9 ± 38.9	23.2 ± 39.1	11.4 ± 36.3	0.295
	CEA (ng/mL)	12.8 ± 131.8	14.1 ± 140.7	4.3 ± 7.4	0.534
	CA-125 (U/mL)	65.2 ± 201.5	63.1 ± 227.8	70.4 ± 116	0.650
LN metastasis	Negative	282 (70.9)	250 (70.6)	32 (72.7)	0.772
	Positive	116 (29.1)	104 (29.4)	12 (27.3)	
Tumor size (cm)	Mean ± SD	5.3 ± 1.9	5.4 ± 1.8	5.4 ± 2.3	0.888
	≤4.0	97 (24.4)	84 (23.7)	13 (29.5)	0.397
	>4.0	301 (75.6)	270 (76.3)	31 (70.5)	
Duration of RT (days)	Median (IQR)	54 (49–60)	54 (49–60)	53 (49–58)	0.350

Clinicopathological characteristics of patients with SCC or AC/ASC. SCC, squamous cell carcinoma; AC, adenocarcinoma; ASC, adenosquamous carcinoma; Hb, hemoglobin; LN, lymph node; SCC., squamous cell carcinoma antigen; CEA, carcinoembryonic antigen; CA-125, cancer antigen-125; FIGO, International Federation of Gynecology and Obstetrics; SD, standard deviation; IQR, interquartile range; RT, radiation therapy.

**Table 2 jcm-09-03471-t002:** Treatment response according to histologic subtype and radiotherapy phase during concurrent chemoradiotherapy (CCRT).

RT Response	SCC	AC/ASC	*p*-Value
Post-EBRT response			
EBRT_regression_ (%)	53.6% ± 26.1	32.8% ± 29.9	<0.001
CR	48 (13.6)	3 (6.8)	<0.001
PR	245 (69.2)	18 (40.9)	
SD	61 (17.2)	23 (52.3)	
PD	0	0	
Post-ICR response			
ICR_regression_ (%)	73.6% ± 31.3	77.9% ± 22.7	0.400
CR	163 (53.3)	19 (46.3)	0.683
PR	107 (35.0)	17 (41.5)	
SD	36 (10.2)	5 (12.2)	
PD	0	0	
Post-RT response			
RT_regression_ (%)	87.0% ± 19.3	83.2% ± 22.3	0.222
CR	211 (59.6)	22 (50.0)	0.474
PR	136 (38.4)	21 (47.7)	
SD	7 (2.0)	1 (2.3)	
PD	0	0	
EBRT_proportion_ (%)	60.5% (43.1–79.5)	30.9% (13.7–56.6)	<0.001
Good EBRT responders (EBRT_regression_ > 26%)	303 (85.6)	23 (52.3)	<0.001
Poor EBRT responders (EBRT_regression_ ≤ 26%)	51 (14.4)	21 (47.7)	
More EBRT responders (EBRT_proportion_ > 40%)	277 (78.2)	17 (38.6)	<0.001
More ICR responders (EBRT_proportion_ ≤ 40%)	77 (21.8)	27 (61.4)	
Good RT responders (RT_regression_ > 92%)	212 (50.9)	24 (54.5)	0.518
Poor RT responders (RT_regression_ ≤ 92%)	142 (40.1)	20 (45.5)	

RT, radiation therapy; SCC, squamous cell carcinoma; AC, adenocarcinoma; ASC, adenosquamous carcinoma; EBRT, external beam radiotherapy; ICR, intracavitary radiotherapy; EBRT _regression_, post-EBRT regression rate; ICR _regression_, post-ICR regression rate; RT _regression_, overall regression rate; EBRT _proportion_, proportion of tumor regression after EBRT to overall regression; CR, complete response; PR, partial response; PD, progressive disease; SD, stable disease.

**Table 3 jcm-09-03471-t003:** Multivariate analysis of prognostic factors for PFS/OS.

Characteristics	PFS		OS	
	Hazard Ratio (95%, CI)	*p*-Value	Hazard Ratio (95%, CI)	*p*-Value
Cell type				
SCC	1		1	
AC/ASC	2.37 (1.45–3.89)	0.001	1.91 (1.18–3.09)	0.009
Pretreatment Hb				
<11 (g/dL)	1		1	
≥11 (g/dL)	1.07 (0.71–1.64)	0.727	0.92 (0.62–1.35)	0.687
Stage				
II	1			
III	1.52 (0.99–2.33)	0.056	1.91 (1.19–3.05)	0.007
IV	1.47 (0.79–2.73)	0.219	2.08 (1.05–4.12)	0.035
LN metastasis				
Negative	1		1	
Positive	1.12 (0.75–1.66)	0.890	1.05 (0.67–1.32)	0.253
Tumor size				
≤4.0 cm	1		1	
>4.0 cm	1.64 (1.01–2.67)	0.044	1.52 (0.95–2.42)	0.037
EBRT _regression_				
>26%	1		1	
≤26%	2.16 (1.38–3.37)	0.001	2.53 (1.54–3.67)	0.008
EBRT _proportion_				
>40%	1		1	
≤40%	2.43 (1.12–2.56)	0.031	2.59 (1.15–2.74)	0.015

PFS, progression-free survival; OS, overall survival, SCC, squamous cell carcinoma; AC, adenocarcinoma; ASC, adenosquamous carcinoma; Hb, hemoglobin; LN, lymph node; EBRT, external beam radiotherapy; EBRT_regression_, post-EBRT tumor size regression rate; EBRT_proportion_, proportion of EBRT to overall size regression.

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
