# Peer review of "Prognostic Significance of Tumor Regression Rate during Concurrent Chemoradiotherapy in Locally Advanced Cervix Cancer: Analysis by Radiation Phase and Histologic Type"

_jcm, 2020, doi:10.3390/jcm9113471_

Round 1

Reviewer 1 Report

Good paper on a topic that well all wish there was more data. It has the limitations of any retrospective study and I hope the authors develop a prospective study to address the same issue

Author Response

We thank the reviewer for positive comments. We also agree on the limitations of retrospective design like our study. We are planning a prospective study on this topic and look forward to submitting our findings to this journal in the future.

Reviewer 2 Report

The authors ask two interesting questions in this study: 1) if there is a difference in outcomes between patients with squamous cell carcinoma of the cervix versus those with adenocarcinoma or adenosquamous carcinoma and 2) if there is a difference in outcome between those patients that have a good response to EBRT versus those that had a poor response to EBRT.  There are some concerns which need to be addressed.

  • The authors do clearly discuss the controversy regarding the prognosis of adenocarcinoma versus squamous cell carcinoma. Another important study to reference and discuss in the Discussion section is the SEER study by Galic V et al., Gynecologic Oncology, 2012; 125(2): 287-291.  This study included over 24,000 cervix cancer patients and showed that for both early and advanced stage diseases, women with adenocarcinoma had an increased likelihood of dying from disease compared to those with squamous neoplasms (HR 1.39 & HR 1.21, respectively).
  • In section 2.1 (Inclusion and Criteria) in the second paragraph, the authors state, “EBRT was delivered to the whole pelvis five times per week with a 10 MV photon beam at a daily dose of 1.8 Gy, for a total dose of 50.4 Gy over the entire course of treatment.” However, in the sentences before and after this statement, the authors state an EBRT dose of 41.4 to 45 Gy was given.  Please clarify.
  • Please describe how EBRT was given. With IMRT? With 4-field box technique?
  • The authors state that intracavitary brachytherapy was used. Were any patients treated with interstitial brachytherapy?
  • A major concern with this study is the radiation dose prescription used. The brachytherapy dose was 6 fractions of 4 Gy each.  Please provide a reference for this dose fractionation since this is not a standard prescription dose for brachytherapy.  The standard recommended total dose for cervix cancer (EBRT + brachytherapy) is 80-90 Gy EQD2.  However, with the dose fractionation used in this study (1.8 Gy x 25 fractions EBRT + 4 Gy x 6 fractions brachytherapy) the total dose is only 72.2 Gy EQD2.  Why was such a low dose prescription used?  Also, the authors state that the brachytherapy dose was prescribed at point; is this in reference to point A based 2-dimensional planning?  Additionally, since MRI scans were obtained at least 3 times throughout the management of the patients, why was MRI-based brachytherapy or CT-based brachytherapy planning not used for any patient? 
  • In Table 1, the authors list the clinicopathological characteristics of the patients in this study. Please list the tumor grade of the patients in the study since grade is also known to be a prognostic factor.  In Table 3, please add grade as a factor in the multivariate analysis.

Reviewer 3 Report

This is a well written manuscript. The study is clearly presented and the design seems appropriate. 

The authors recognize the limitations of their study but they report interesting findings which others will be able to confirm in larger cohort. 

I only have few minor comments that should be taken into consideration before approving the manuscript.

1) can the authors add the number of patients in the statistical analysis presented int he result section? For example, line 140: (53.6% vs 32.8%, p<0.001, N=?). 

2) In table 1. Can the authors also present the demographic data of patients with  know prognostic factors such as, tutor grade, anemia, lymph node status, deep stroll invasion and perineurial invasion. What chemotherapy drugs were used ?

3) Can the authors also include these parameters in the univariate analysis and include the significant ones in the multivariate analysis ?

Round 2

Reviewer 2 Report

Accept in present form.